Assessment of physical education teachers’ use of distance teaching behavior under the influence of the COVID-19 pandemic

Huang Hsiu-Chin 1
Kung Ya-Tzu 2
Huang Ruey-Rong 3
Mui Wui-Chiu 4 yamemui@yahoo.com
Su Yu-Chien 5
1 School of Physical Education and Health, Zhaoqing University , Guangdong , China
2 Office of Physical Activities, National Pingtung University , Pingtung , Taiwan
3 Health Science and Management , Chung Jen Junior College of Nursing , Taiwan
4 Department of Anesthesiology, Ditmanson Medical Foundation Chia-Yi Christian Hospital , Chiayi , Taiwan
5 Department of Physical Education, Health & Recreation, National Chiayi University , Chiayi , Taiwan
Khoo Selina
Electronic publication date: 2025 Jan 20
Publication date: 2025
Volume: 13
Electronic Location ID: e18743
Received 2024 Jun 19; Accepted 2024 Dec 2
Copyright: © 2025 Huang et al.
Copyright year: 2025
Copyright holder: Huang et al.
License: This is an open access article distributed under the terms of the Creative Commons Attribution License, which permits unrestricted use, distribution, reproduction and adaptation in any medium and for any purpose provided that it is properly attributed. For attribution, the original author(s), title, publication source (PeerJ) and either DOI or URL of the article must be cited.
License URL: https://creativecommons.org/licenses/by/4.0/

Keywords: UTAUT, Distance teaching, Behavioral intention, Use behavior

Funding: The authors received no funding for this work. The funders had no role in study design, data collection and analysis, decision to publish, or preparation of the manuscript The authors received no funding for this work.

==============================
Background

For many physical education teachers, being suddenly forced to switch from traditional face-to-face teaching to online teaching without adequate mental preparation posed numerous challenges and difficulties. Therefore, the purpose of this study is to validate the use of distance teaching behavior models for physical education teachers under the influence of the COVID-19 pandemic.

Method

The unified theory of acceptance and use of technology (UTAUT) model was employed to explore the use intention and use behavior of distance teaching. The model contains four independent variables: performance expectancy (PE), effort expectancy (EE), social influence (SI), and facilitating conditions (FC), two dependent variables: behavioral intention (BI) and use behavior (UB) and three moderating variables: gender, age, and experience. A total of 400 questionnaires were distributed to elementary and junior high school physical education teachers with 364 valid responses. Partial least squares structural equation modeling (PLS-SEM) was used to test the relationships among variables.

Results

The results found PE, EE and SI had significant influences on BI and FC and BI had significant influences on UB. Experience had moderating effects among SI and BI.

Conclusions

Based on the results, recommendations for physical education teachers and schools are proposed. Furthermore, research limitations and future directions are discussed.

Introduction

In 2021, the World Health Organization (WHO) announced the discovery of a new virus, Coronavirus Disease-2019 (COVID-19), now commonly known as the coronavirus. Based on weekly epidemiological updates on confirmed COVID-19 cases, the number of confirmed cases for this global pandemic had reached 206 million people by the end of 2021, requiring just half a year to increase from 100 million confirmed cases (World Health Organization, 2021).

Taiwan was first affected by the epidemic in early 2020. At the beginning of the outbreak, the Ministry of Education urgently ordered a 2-week postponement of the start of the school year and began to deploy online teaching mechanisms. Until May 18, 2021, as the epidemic intensified, the Ministry of Education announced the nationwide suspension of all levels of schools, switching to online learning at home (Lin & Chen, 2022). Because of its extreme health risks and its rapid rate of transmission, the COVID-19 crisis compelled schools in Taiwan adopt distance learning education.

For physical education, being suddenly forced to switch from established in-person classes to novel distance learning via online instruction posed unique challenges to teachers and students alike. Challenges included teacher acceptance and their competence to use a new technology for instructing students in a subject area that has traditionally required in-person teaching. Therefore, to assess the success of this mandated distance-learning physical education program, this study explored the behavior of physical education teachers and their technology adoption and use by applying the unified theory of acceptance and use of technology (UTAUT) proposed by Venkatesh et al. (2003).

The technology integration method applied in this study is based on developing an empirical model based on the possible factors influencing teachers’ intentions and behaviors in distance teaching. The goal of this research is to provide results and recommendations for relevant institutions and individuals for increasing the proficiency of physical education teachers in distance teaching.

UTAUT was proposed by Venkatesh et al. (2003), who compared and integrated different information technology acceptance models to understand users’ use intentions. The research model consists of four key factors that influence users’ behavioral intentions and use behavior, namely, performance expectancy, effort expectancy, social influence and facilitating conditions. Performance expectancy is defined as the degree to which users believe that using the information system will help them accomplish tasks, indicating the extent to which individuals perceive that using the system will assist in achieving performance goals. Effort expectancy is defined as the ease of use of the information system, representing the ease or difficulty users perceive in operating the technology. Social influence is defined as the extent to which users perceive that others believe they should use the information system, indicating the degree to which individuals are influenced by their surrounding social groups. Facilitating conditions is defined as the resources provided by hardware facilities and technical systems, representing the degree of assistance provided by software and hardware resources. In addition, the model also includes four moderating variables: gender, age, experience, and voluntariness of use which are posited to moderate the impact of the four key constructs on use intention and behavior.

Since the introduction of the UTAUT model in 2003, it has been widely applied in various research fields. Whether in theoretical foundations or empirical studies, it is a highly comprehensive model. Many researchers had conducted studies based on its theory, accumulating considerable research data in both theoretical and practical aspects. For example, the UTAUT model has been applied in a wide range of fields, including business, trade, healthcare, social media, system software, food shopping, educational assistance, artificial intelligence systems, and more (Aydin, 2023; Budhathoki et al., 2024; Chang et al., 2022; Namahoot & Jantasri, 2023; Raffaghelli et al., 2022; Sudirjo et al., 2023).

In this era of technological advancement, digital technology is rapidly entering the workplace. Many institutions and researchers are trying to integrate various fields with digital technology. In order to do so successfully, industries need to understand users’ willingness to accept and use their products and technologies through the UTAUT model.

Researchers are curious about the current state of distance teaching, particularly among physical education teachers, who have been implementing it for some time now. As introduced above, amidst the rampant spread of the COVID-19 pandemic, teachers across Taiwan have embarked on the path of distance teaching under policy initiatives. Apart from academic subjects, practical subjects such as health and physical education, arts and humanities, and integrated curriculum often require direct physical intervention for instruction (e.g., properly throwing a baseball). This may present challenges in distance teaching due to the need for teachers to personally demonstrate and assist. Given the enforcement of distance teaching, what are their intentions regarding distance teaching in the classroom? Do these intentions affect their actual use of distance teaching methods? Therefore, this study will employ the technology integration model (UTAUT) to verify the behavioral intention and use behavior of physical education teachers in distance teaching, providing governmental departments and relevant units with reference points for promoting distance education.

UTAUT encompasses six constructs, which are performance expectancy, effort expectancy, social influence, facilitating conditions, behavioral intention and use behavior. In this model, the first four constructs are independent variables and the latter two are dependent variables. UTAUT also uses age, gender and experience as moderators. The next section will introduce the relationships of the variables based on the UTAUT model derived from previous research and present our research hypotheses developed during this study.

Relationship between performance expectancy and behavioral intention

The term “performance” originates from economics, referring to the results achieved from work, which is influenced by knowledge and skills (Stahel, 2010). As early as 2003, Venkatesh et al. (2003) developed this concept into performance expectancy and applied it to people’s acceptance of technology systems. When people have higher performance expectations for a system, they are more likely to enhance their behavioral intentions. In a study on individuals’ decisions to share experiential content on social networking sites (SNS), Herrero-Crespo, Gutiérrez & Garcia-De (2017) found that performance expectancy significantly influences behavioral intention.

When people engage in learning through electronic systems, whether in developed or developing countries, performance expectancy remains an important predictor of behavioral intention (El-Masri & Tarhini, 2017). Similarly, in a study on students’ intention to use Moodle, it was found that performance expectancy influences students’ behavioral intention to use Moodle (Aydin, 2023). Therefore, this study proposes the following hypothesis:

H1: Performance expectancy has a significant positive effect on teachers’ behavioral intention to engage in distance teaching.

Relationship between effort expectancy and behavioral intention

When an individual or an institution is confronted with the prospect of using an innovative technology, its ease of operation is often an important consideration for its adoption. Rogers (1983) proposed the innovation diffusion theory (IDT), measuring user acceptance with complexity. Both the technology acceptance model (TAM) and the model of PC utilization (MPCU) indicate that perceived ease of use affects users’ willingness to use technology products. When users perceive technology products as easy to use, it enhances their behavioral intention (Davis, 1989; Thompson, Higgins & Howell, 1991).

Venkatesh et al. (2003) combined these concepts to develop the UTAUT model, introducing the concept of effort expectancy to explore the ease of use of technology systems. For example, Chang et al. (2019) found that effort expectancy positively influences users’ intention to use online hotel booking. Similarly, Azizi, Roozbahani & Khatony (2020) found that effort expectancy significantly positively influences medical students’ intention to use blended learning (combining face-to-face and online learning). Chang et al. (2022) also found that effort expectancy positively influences university sports teachers’ intention to use social networking sites. Therefore, this study proposes the following hypothesis:

H2: Effort expectancy has a significant positive effect on teachers’ behavioral intention to engage in distance teaching.

Relationship between social influence and behavioral intention

The concept of social influence has been proposed in both the theory of reasoned action (TRA) and the theory of planned behavior (TPB). Both theories assert that personal attitude and subjective norms influence people’s behavioral intentions, where subjective norms refer to the thoughts and attitudes of significant others (Fishbein & Ajzen, 1975; Ajzen, 1991). Moore & Benbasat (1991) also suggest that in IDT, behavioral intention is influenced by image; users believe that enhancing social status and image within a group will affect behavioral intention. This concept aligns with the notion of social influence introduced by Venkatesh et al. (2003). Budhathoki et al. (2024) found that students’ social influence significantly affects their intention to use ChatGPT. Hsu (2023) discovered in a study on students’ use of Language Massive Open Online Courses (LMOOCs), that social influence affects their use of LMOOCs. Chang et al. (2019) indicated that social influence positively affects customers’ intention to use online hotel booking. Azizi, Roozbahani & Khatony (2020) in their empirical study suggested that providing important social resources and organizational support are crucial steps for successful implementation of blended learning systems by students. Therefore, the study proposes the following hypothesis:

H3: Social influence has a significant positive effect on teachers’ behavioral intention to engage in distance teaching.

Relationship between facilitating conditions, behavioral intention, and use behavior

The term “facilitating conditions” refers to the extent to which users perceive that relevant organizational units can provide support, such as hardware and software equipment, and technical assistance (Venkatesh, Thong & Xu, 2012). Thompson, Higgins & Howell (1991) suggested that having sufficient resources and assistance directly enhances the use behavior of a particular technology. The TPB and the combined theory of planned behavior and technology acceptance model (C-TAM-TPB) both mentioned that behavioral intention is not the only pathway to actual use behavior; perceived behavioral control not only influences behavioral intention but also directly affects use behavior. Even with behavioral intention, if users perceive a lack of control, they will not engage in use behavior (Ajzen, 1985). Venkatesh et al. (2003) argued that facilitating conditions significantly impact both behavioral intention and use behavior.

For example, Chang et al. (2019) found that facilitating conditions positively affected customers’ intention to use online hotel booking. In a study by Jahanshahi, Tabibi & Van Wee, 2020 using the UTAUT model to investigate people’s intention and use behavior regarding bike-sharing systems, facilitating conditions emerged as the sole significant factor influencing use behavior. Chang et al. (2022) discovered that facilitating conditions positively influence the intention of university sports teachers to use SNS. Therefore, this study proposes the following hypothesis:

H4: Facilitating conditions have a significant positive effect on teachers’ use behavior in distance teaching.

Relationship between behavioral intention and use behavior

Behavioral intention refers to the degree to which an individual intends to engage in a particular behavior and is a psychological attitude preceding human action, while use behavior refers to actual behavior (Liu et al., 2016). Human behavior stems from behavioral intention, and the more positive the psychological attitude, the more likely that use behavior will occur (Fishbein & Ajzen, 1975). In the past, many researchers had considered behavioral intention as a crucial indicator for predicting behavior (Chang et al., 2022; Budhathoki et al., 2024; Liu et al., 2016; Tian & Yang, 2023). Liu et al. (2016) found in their study on the use behavior of SNS among university physical education students that higher behavioral intention leads to more frequent use behavior among university students. Chang et al. (2022) also found that the intention of university physical education teachers to use SNS positively influences their use behavior. Therefore, the hypothesis is proposed as follows:

H5: The behavioral intention of teachers to engage in distance teaching behavior has a significant positive effect on their use behavior.

Gender, age, and experience as moderators in the UTAUT model

Life course theory suggests that age has a significant impact on human growth and development. Different generations, shaped by different historical and cultural contexts, are influenced differently by cognitive, behavioral, and environmental factors. As individuals age, those within the same generation tend to be assigned similar tasks and expectations by society. Thus, age strata significantly influence individuals’ autonomy and choices. In addition to age, gender also exerts a similar influence. Gender schema theory posits that gender is profoundly influenced by social construction, with societal expectations differing for males and females from childhood onward. Hence, gender and age both play pivotal roles in the social development process. Venkatesh et al. (2003) mentioned in their UTAUT study that gender, age, and experience all have moderating effects on performance expectancy, effort expectancy, social influence, and facilitating conditions. Chen (2017) pointed out that teachers’ intention to use innovative information technology is influenced by gender and age. Chang et al. (2019) found that gender, age, and experience moderate the use behavior in the online hotel booking model. Moreover, Chang et al. (2022) discovered that gender, age, and experience moderate students’ behavior in using social networking sites. Combining the aforementioned findings, this study incorporates gender, age, and experience into the research model, proposing the following three sets of hypotheses:

(1) Moderating effect of gender

H6a: Gender moderates the effect of performance expectancy on teachers’ intention to engage in distance teaching.

H6b: Gender moderates the effect of effort expectancy on teachers’ intention to engage in distance teaching.

H6c: Gender moderates the effect of social influence on teachers’ intention to engage in distance teaching.

H6d: Gender moderates the effect of facilitating conditions on teachers’ behavioral intention to engage in distance teaching behavior.

(2) Moderating effect of age

H7a: Age moderates the effect of performance expectancy on teachers’ intention to engage in distance teaching.

H7b: Age moderates the effect of effort expectancy on teachers’ intention to engage in distance teaching.

H7c: Age moderates the effect of social influence on teachers’ intention to engage in distance teaching.

H7d: Age moderates the effect of facilitating conditions on teachers’ behavioral intention to engage in distance teaching.

(3) Moderating effect of experience

H8a: Experience moderates the effect of performance expectancy on teachers’ intention to engage in distance teaching.

H8b: Experience moderates the effect of effort expectancy on teachers’ intention to engage in distance teaching.

H8c: Experience moderates the effect of social influence on teachers’ intention to engage in distance teaching.

H8d: Experience moderates the effect of facilitating conditions on teachers’ behavioral intention to engage in distance teaching.

In summary, the presented research framework is comprised of four independent variables, three moderating variables, and two dependent variables. The independent variables include performance expectancy, effort expectancy, social influence, and facilitating conditions. The moderating variables consist of gender, age, and experience. The dependent variables are behavioral intention and use behavior. The research framework is presented in Fig. 1.

Figure 1 The hypothesized conceptual model of the study.

Materials and methods

Participants

Participants in this study were selected from elementary school and junior high school physical education teachers in six municipalities in Taiwan who engaged in distance teaching. The survey was approved by National Chiayi University, Taiwan, and the surveys were conducted over a two month period from March 1 through April 30, 2022. The six municipalities are located in north, central and south Taiwan. The combined population of the cities is more than 2.5 million people, and the physical education classes were taught by teachers who hold a physical education degree. In other areas, some physical education classes are often taught by teachers without physical education degree. Since the study focused on physical education teachers, the physical education teachers in municipalities were selected to accurately reflect the trained profession. To initiate the survey, the principals of selected schools received invitations from the research team, and if they agreed to participate in the survey, the research team distributed the questionnaires and then collected the responses. Written informed consent was received. A total of 400 questionnaires were distributed and 364 valid responses were obtained with a valid return rate of 91%. The raw data aligns with the journal’s data-sharing policy and can be accessed for further research.

Research instruments

Since the study employed the UTAUT model (Venkatesh et al., 2003), the original questionnaire was modified to conform with the physical education scenario. In short, the research instruments consisted of four parts. The first part of the scale consists of collecting basic information about the subjects. The second part, based on the model questions designed by Venkatesh et al. (2003), was modified and constructed by the researcher. For this study, it consisted of a total of 24 questions. Responses to all questions utilized a 7-point Likert Scale, including options such as “strongly agree,” “agree,” “somewhat agree,” “neutral,” “disagree,” “somewhat disagree,” and “strongly disagree” for measurement.

Due to the extensive empirical research foundation and established reliability and validity of the UTAUT model scale, it represents a fairly comprehensive questionnaire model. The researcher referenced relevant scales from Venkatesh et al. (2003), Chang et al. (2019, 2022), among others, and modified them according to the research content and subjects of this study. After consulting three scholars and experts, the questionnaire was refined, and operational definitions of each variable were redefined. The item descriptions were then consolidated as Table 1.

Table 1 Items of teaching behavior scale.

Variable	Operational definition	Item	Source	
Performance expectancy	The extent to which teachers perceive that system operation can assist in carrying out distance teaching activities.	1) During the pandemic, I can enhance students’ interest in physical education classes through distance teaching.

2) During the pandemic, I can guide students in learning physical education through distance teaching.

3) During the pandemic, I can design physical education courses using distance teaching methods.

4) During the pandemic, distance teaching can improve my performance in physical education classes.

	Venkatesh et al. (2003)
Herrero-Crespo, Gutiérrez & Garcia-De (2017)
Chang et al. (2022)	
Effort expectancy	The perceived complexity and difficulty of system operation when teachers engage in distance teaching behavior.	1) During the pandemic, I do not find it complicated to deliver physical education classes through distance teaching.

2) I am very familiar with delivering physical education classes through distance teaching methods.

3) I have enough knowledge about the system and functionalities of distance teaching.

4) Engaging in physical education classes through distance teaching is easy for me.

	Venkatesh et al. (2003)
Chang et al. (2022)	
Social influence	The extent to which teachers are influenced by significant social groups when engaging in distance teaching behavior, affecting their decision to use it or not.	1) I use distance teaching for physical education classes because people around me do the same.

2) The school expects me to conduct physical education classes through distance teaching.

3) As more and more teachers start using distance teaching for physical education classes, I will also follow suit.

4) If newspapers, magazines, and news media mention that distance teaching is a future trend, I will try using it for physical education classes.

	Venkatesh et al. (2003)
Venkatesh, Thong & Xu, (2012)
Chang et al. (2022)	
Facilitating conditions	The extent to which teachers can access software and hardware resources when engaging in distance teaching behavior.	1) I have sufficient hardware equipment (internet, computer) for distance teaching.

2) I can learn how to conduct physical education classes through distance teaching via workshops and training sessions.

3) Whenever I encounter difficulties in distance teaching, there is always someone available to support me and help me solve problems.

4) I often feel frustrating in operating software and hardware equipment during distance teaching.

	Venkatesh et al. (2003)
Chang et al. (2022)	
Behavior intention	Teachers’ attitudes and willingness to use distance teaching behavior.	1) Due to the influence of the pandemic, I believe that conducting physical education classes through distance teaching is an efficient teaching method.

2) Using distance teaching for physical education classes is one of the teaching methods I would choose in the future.

3) After the outbreak of the pandemic, I am willing to learn and try conducting physical education classes through distance teaching.

4) I believe it is necessary to learn how to conduct physical education classes through distance teaching.

	Venkatesh et al. (2003)
Chang et al. (2019,
2022)	
Use behavior	The frequency teachers have actually engaged in distance teaching behavior over the past year.	1) Due to the impact of the pandemic, I started conducting physical education classes through distance teaching.

2) Due to the impact of the pandemic, I will engage students in physical activities through online video sessions or by recording videos.

3) After the outbreak of the pandemic, I conducted physical education classes through distance teaching very frequently.

4) After the outbreak of the pandemic, I have never conducted physical education classes through distance teaching.

	Chang et al. (2019,
2022)	

To ensure the quality of research instrument, common method variance test, multicollinearity diagnostics, tests of reliability and validity were performed.

Data analysis

The collected raw data was edited and examined to delete incomplete or erroneous records. Following this quality control, the data was analyzed using two statistical software packages. SPSS 26 was used to perform basic statistical analyses which included descriptive statistics of the participants. For the UTAUT model, the eight hypotheses were tested using partial least squares (PLS), a regression method for studying complex multivariate relationships among observed and latent variables. Similar to structural equation modeling (SEM), PLS measures the correlation of constructs and analyzes the reliability and validity of each construct (Chin, 1998). PLS is a popular statistical method in the information discipline and is common in UTAUT and UTAUT2 studies (Venkatesh et al., 2003; Venkatesh, Thong & Xu, 2012). One advantage of PLS among other SEM tools is the outstanding ability to estimate moderating effects simultaneously along with direct impacts of independent latent variables. Especially in this study, 12 moderating effects were to be explored. Therefore, PLS was chosen for hypothesis validation. There are many statistical software packages that enable users to perform PLS analysis (e.g., SmartPLS, Excel, and Warp PLS). Warp PLS 8.0 developed by Kock (2022) was chosen for the present study.

Results

Descriptive statistics of the respondents

Table 2 present the descriptive statistics of the respondents. In terms of gender, there were 165 males (45.3%) and 199 females (54.7%). Regarding teaching category, there were 143 junior high school teachers (39.3%) and 221 elementary school teachers (60.7%). As for education level, there were 157 participants with bachelor’s degrees (43.1%), 200 with master’s degrees (54.9%), and seven with doctoral degrees (1.9%). In terms of distance teaching experience, 22 participants (6.0%) were unfamiliar, 293 (80.5%) were moderately familiar, and 49 (13.5%) were quite familiar. The average age of the 364 participants was 36.69 years (standard deviation 8.36). The youngest teacher was 23 years old, and the oldest was 63 years old. The average teaching experience was 11.02 years (standard deviation 7.58), ranging from 1 to 37 years. The average experience in distance teaching was 6.17 months (standard deviation 5.94), ranging from 0 to 30 months.

Table 2 Descriptive statistics of respondent (N = 364).

Variables	Category	N	Frequency(%)	
Sex	Male	165	45.3	
Female	199	54.7	
Teaching level	Elementary school	221	60.7	
Junior high school	143	39.3	
Education	Undergraduate	157	43.1	
Master	200	54.9	
PhD	7	1.9	
Experience in distance teaching	Not familiar	22	6.0	
Somewhat familiar	293	80.5	
Very familiar	49	13.5	
Location	Taipei City	58	15.9	
New Taipei City	53	14.6	
Taoyuan City	52	14.3	
Taichung City	66	18.1	
Tainan City	62	17.0	
Kaohsuing City	73	20.1	

Apps used for distance teaching were investigated using multiple-choice questions. There were a total of 446 responses on the use of distance teaching apps. Ranked by proportion, the most used was GT with 322 responses (72.2%), followed by YouTube with 60 responses (13.5%), and the lowest used was Jisit with three responses (0.7%) (see Table 3).

Table 3 Statistics of apps used in distance teaching (multiple choice, N = 364, grand total = 446).

App	Response	Percentage of observations	
N	Frequency (%)	
GT	322	72.2	88.5	
MS	38	8.5	10.4	
YouTube	60	13.5	16.5	
Line	6	1.3	1.6	
Jisit	3	0.7	0.8	
Others	17	3.8	4.7	
Notes:

Frequency (%) = N/446 × 100%.

Percentage of observations = N/364 × 100%

Reliability and validity of the research instrument

1. Common Method Variance and Collinearity Test

Avolio, Yammarino & Bass (1991) pointed out that self-reported questionnaire surveys, commonly used in research, may lead to single source bias due to data being collected from the same group of respondents, resulting in common method variance (CMV). In the model of this study, the six latent variables are prone to common method variance. Podsakoff et al. (2003) suggested using exploratory factor analysis by setting all items as one factor to examine the unrotated factor analysis results, known as Harman’s one-factor test. When the explanatory power of one factor extracted from factor analysis does not exceed 50%, it indicates that there is no significant common method bias in the research sample (Mossholder et al., 1998; Podsakoff & Organ, 1986). In this study, all items were subjected to factor analysis together, and the explanatory variance of one factor extracted was 43.59%, which is not greater than 50%, indicating no significant common method variance issue.

2. Collinearity Test

For the study model, a collinearity test was conducted for the six latent variables. According to Hair et al. (2014) and Kock & Lynn (2012), the general loose criterion is varience inflation factor (VIF) < 10, the strict criterion is VIF < 5, and the most stringent criterion is VIF < 3.3. Based on the analysis results using Warp PLS 8.0 statistical software, the VIF values of the six latent variables in this study model ranged from 1.79 to 3.10. Therefore, the VIF values of the six latent variables in this study model are all below the criterion of 3.3, meeting the most stringent criterion, indicating that there is no serious collinearity problem.

3. Reliability Analysis

According to Fornell & Larcker (1981), when the composite reliability (CR) and Cronbach’s α reliability coefficient are equal to or greater than 0.70, it indicates that all observed variables within the latent variable have internal consistency reliability. Table 4 presents the results of the composite reliability and Cronbach’s α values for each variable in the model of this study. The composite reliability value for performance expectancy is 0.92, with a Cronbach’s α value of 0.88; for effort expectancy, the composite reliability value is 0.92, with a Cronbach’s α value of 0.89; for social influence, the composite reliability value is 0.84, with a Cronbach’s α value of 0.74; for facilitating conditions, the composite reliability value is 0.87, with a Cronbach’s α value of 0.80; for behavioral intention, the composite reliability value is 0.88, with a Cronbach’s α value of 0.81; and for use behavior, the composite reliability value is 0.87, with a Cronbach’s α value of 0.79. Overall, the composite reliability values and Cronbach’s α values for each latent variable are above 0.70, meeting the testing standards, indicating that the reliability of the latent variables in this study were acceptable.

Table 4 Reliability test.

Variables	Composite reliability	Cronbach’s α	
Performance expectancy (PE)	0.92	0.88	
Effort expectancy (EE)	0.92	0.89	
Social influence (SI)	0.84	0.74	
Facilitating conditions (FC)	0.87	0.80	
Behavioral intention (BI)	0.88	0.81	
Use behavior (UB)	0.87	0.79	

4. Convergent Validity

Convergent validity refers to whether the factor loadings of measurement variables on their latent variables are sufficiently large. It is generally recommended that factor loadings should be greater than 0.50 (Hair et al., 2009). Table 5 presents the factor loadings of 24 measurement variables for the six latent variables in the model of this study, ranging from 0.64 to 0.90, all of which are greater than 0.50, thus meeting the recommendation by Hair et al. (2009). The test results indicate that all six latent variables in this study demonstrate good convergent validity.

Table 5 Factor loading of each indicator to its corresponding construct.

Latent variable	Indicator	Factor loading	
Performance expectancy (PE)	PE1	0.85	
	PE2	0.87	
	PE3	0.84	
	PE4	0.87	
Effort expectancy (EE)	EE1	0.87	
	EE2	0.90	
	EE3	0.79	
	EE4	0.89	
Social influence (SI)	SI1	0.66	
	SI2	0.71	
	SI3	0.87	
	SI4	0.76	
Facilitating condition (FC)	FC1	0.83	
	FC2	0.73	
	FC3	0.80	
	FC4	0.82	
Behavioral intention (BI)	BI1	0.82	
	BI2	0.87	
	BI3	0.68	
	BI4	0.84	
Use behavior (UB)	UB1	0.85	
	UB2	0.83	
	UB3	0.81	
	UB4	0.64	

5. Discriminant Validity

Discriminant validity primarily assesses whether there is differentiation among all latent variables in the research model. Chin (1998) proposed that the square root of the average variance extracted (AVE) for each individual construct should be greater than or equal to 0.50, and it should be greater than the covariance between that construct and other constructs in the model. Table 6 presents the square roots of the AVE for the six latent variables in the study model, ranging from 0.75 to 0.86, all of which are greater than all correlation coefficients within the same column and row, and they all exceed 0.50, meeting the recommended standard. Therefore, the six latent variables in this study demonstrate good discriminant validity.

Table 6 Discriminant validity of construct.

Construct	Correlation / averagevarianceextracted	
(1)	(2)	(3)	(4)	(5)	(6)	
PE(1)	0.86a						
EE(2)	0.73	0.86a					
SI(3)	0.53	0.50	0.75a				
FC(4)	0.52	0.70	0.53	0.80a			
BI(5)	0.69	0.64	0.59	0.60	0.80a		
UB(6)	0.52	0.54	0.55	0.52	0.58	0.79a	
Note:

PE: performance expectancy; EE: effort expectancy; SI: social influence; FC: facilitating conditions; BI: behavioral intention; UB: use behavior; a: square root of average variance extracted.

Based on the above analyses, the research model has passed tests for common method variance and collinearity among latent variables. The reliability and validity analyses of the measurement model for the six latent variables indicate that the CR and Cronbach’s α reliability values meet the reliability requirements. In terms of validity, the six latent variables have passed tests for convergent validity and discriminant validity. Therefore, this study can proceed with further analysis of the hypothesis testing to examine the relationships of influence.

Hypothesis testing

Following the validity and reliability tests, the researchers used WarpPLS 8.0 to test the research hypotheses. Figure 2 shows the path coefficients and significances obtained from WarpPLS. The test results are explained as follows:

Figure 2 SEM results of the standardized model parameter estimation.

Note: “--” represent “path coefficient was not significant” “-” represent “path coefficient was significant”.

H1: Performance expectancy has a significant positive impact on teachers’ intention to engage in distance teaching behavior (β = 0.37, p < 0.05), indicating that higher performance expectancy among teachers leads to a higher intention to engage in distance teaching behavior. Therefore, the research hypothesis is supported.

H2: Effort expectancy has a significant positive impact on teachers’ intention to engage in distance teaching behavior (β = 0.22, p < 0.05), suggesting that higher effort expectancy among teachers leads to a higher intention to engage in distance teaching behavior. Therefore, the research hypothesis is supported.

H3: Social influence has a significant positive impact on teachers’ intention to engage in distance teaching behavior (β = 0.26, p < 0.05), indicating that higher social influence among teachers leads to a higher intention to engage in distance teaching behavior. Therefore, the research hypothesis is supported.

H4: Facilitating conditions have a significant positive impact on teachers’ use behavior in distance teaching (β = 0.21, p < 0.05), suggesting that higher facilitating conditions among teachers lead to higher use behavior in distance teaching. Therefore, the research hypothesis is supported.

H5: Teachers’ intention to engage in distance teaching behavior has a significant positive impact on their use behavior (β = 0.44, p < 0.05), indicating that higher intention to engage in distance teaching behavior among teachers leads to higher use behavior. Therefore, the research hypothesis is supported.

H6a: Gender does not moderate the relationship between performance expectancy and teachers’ intention to engage in distance teaching behavior (β = −0.02, p > 0.05). Therefore, the research hypothesis is not supported.

H6b: Gender does not moderate the relationship between effort expectancy and teachers’ intention to engage in distance teaching behavior (β = −0.02, p > 0.05). Therefore, the research hypothesis is not supported.

H6c: Gender does not moderate the relationship between social influence and teachers’ intention to engage in distance teaching behavior (β = 0.06, p > 0.05). Therefore, the research hypothesis is not supported.

H6d: Gender does not moderate the relationship between facilitating conditions and teachers’ behavioral intention to engage in distance teaching behavior (β = 0.06, p > 0.05). Therefore, the research hypothesis is not supported.

H7a: Age does not moderate the relationship between performance expectancy and teachers’ intention to engage in distance teaching behavior (β = −0.02, p > 0.05). Therefore, the research hypothesis is not supported.

H7b: Age does not moderate the relationship between effort expectancy and teachers’ intention to engage in distance teaching behavior (β = −0.02, p > 0.05). Therefore, the research hypothesis is not supported.

H7c: Age does not moderate the relationship between social influence and teachers’ intention to engage in distance teaching behavior (β = −0.07, p > 0.05). Therefore, the research hypothesis is not supported.

H7d: Age does not moderate the relationship between facilitating conditions and teachers’ behavioral intention to engage in distance teaching behavior (β = −0.05, p > 0.05). Therefore, the research hypothesis is not supported.

H8a: Experience does not moderate the relationship between performance expectancy and teachers’ intention to engage in distance teaching behavior (β = −0.00, p > 0.05). Therefore, the research hypothesis is not supported.

H8b: Experience does not moderate the relationship between effort expectancy and teachers’ intention to engage in distance teaching behavior (β = −0.05, p > 0.05). Therefore, the research hypothesis is not supported.

H8c: Experience moderates the relationship between social influence and teachers’ intention to engage in distance teaching behavior (β = −0.09, p < 0.05), indicating that the more experience, the weaker the relationship between social influence and use behavior. Therefore, the research hypothesis is supported.

H8d: Experience moderates the relationship between facilitating conditions and teachers’ behavioral intention to engage in distance teaching behavior (β = −0.09, p < 0.05), indicating that the more experience, the weaker the relationship between facilitating conditions and use behavior. Therefore, the research hypothesis is supported.

Explanatory power

Explanatory power, often referred to as R2, represents the percentage of variance in the dependent variable explained by the independent variables, indicating the predictive ability of the research model. Therefore, a higher R2 value indicates stronger predictive ability (Hwang, 2015). According to the recommendations by Hair et al. (2014), R2 values of 0.25, 0.50, and 0.75 represent weak, moderate, and strong explanatory power, respectively.

As shown in Fig. 2, the circles representing the intention to use (BI) variable and the actual use behavior variable are marked with their respective R2 values. The R2 value for the intention to use (BI) variable is 0.61, indicating moderate explanatory power. This means that variables such as performance expectancy, effort expectancy, social influence, and facilitating conditions can collectively explain 61% of the variance in the intention to use variable. Additionally, the R2 value for the relationship between the intention to use and actual use behavior is 0.40, also indicating moderate explanatory power. This implies that the intention to use, along with facilitating conditions, can collectively explain 40% of the variance in actual use behavior.

Discussion

From H1, it is known that performance expectancy has a significant positive effect on teachers’ intention to conduct distance teaching, which is consistent with the findings of Herrero-Crespo, Gutiérrez & Garcia-De (2017). Performance expectancy continues to influence behavioral intentions. As the post-pandemic era arrives, there may be considerable skepticism about the effectiveness of distance teaching for physical education teachers. People might think that distance teaching cannot monitor whether students are actually exercising, whether their movements are correct, and immediate corrections from teachers are not possible.

In reality, this served as an alternative solution to maintain student participation in physical activity during the pandemic. Therefore, many physical education teachers actively seek effective distance teaching methods in hopes of helping students maintain exercise habits and improve their physical health. Consequently, if teachers recognize that distance teaching can provide effective execution of physical education curriculum, it will significantly influence their intention to use it positively.

From H2, it showed that effort expectancy has a significant positive effect on teachers’ intention to engage in distance teaching. This finding aligns with the research conducted by Azizi, Roozbahani & Khatony (2020), which focused on students’ engagement in blended learning (combining face-to-face and online learning). Similarly, Chang et al. (2022) also found that physical education teachers’ effort expectancy positively influences their intention to use social networking sites. Therefore, distance teaching is a course that significantly tests information literacy and technological skills.

Particularly for middle and primary school teachers, in order to capture and maintain students’ attention in distance teaching classes, many teachers are experimenting with various software tools as teaching materials. Their goal is to stimulate students’ learning motivation. If the software used for distance teaching is intuitive and user-friendly, and teachers have a higher level of effort expectancy, it can significantly enhance the intention to use distance teaching.

From H3, it is evident that social influence has a significant positive effect on teachers’ intention to engage in distance teaching. This finding is supported by the research of Azizi, Roozbahani & Khatony (2020), which also confirms that social support within a community is a significant factor influencing people’s behavioral intentions. After all, humans are social beings deeply influenced by collective consciousness. During the “stop teaching but keep learning” period initiated by the Ministry of Education in Taiwan, digital communities for teachers began to flourish. For example, within a relatively short year, the Taiwan Online Synchronous Teaching Community on Facebook attracted over one hundred thousand group members. This indicates that under the initiation of distance teaching, teachers actively sought community assistance, sharing digital resources, and unconsciously but directly enhancing their intention to engage in distance teaching through social influence.

From H4, it is evident that convenience conditions have a significant positive impact on teachers’ use behavior in distance teaching. This finding is supported by Jahanshahi, Tabibi & Van Wee (2020), who mentioned in their study on bike-sharing systems that convenience conditions can enhance users’ behavioral intentions. Distance teaching tests both teachers’ and students’ software and hardware equipment, rendering sufficient digital resources and equipment relatively important. In the Ministry of Education’s digital learning promotion program, classes participating in the project can apply for technology-assisted learning aids, and students in these classes are provided with one tablet per person to encourage teachers to delve into digital teaching. During the epidemic period, even though classes were suspended, teachers and students could borrow laptops, and through their educational cloud accounts, they had access to more teaching resources. These initiatives are key to promoting teachers’ engagement in distance teaching.

From H5, it is evident that teachers’ intention to implement distance teaching behavior has a significant positive impact on their use behavior. Previous studies have confirmed that intention is key to predicting behavior (Abbad, 2021). Similarly, Liu et al. (2016) found similar results in their study on college students’ social network behavior. From this, it can be inferred that despite engaging in distance teaching under policy development, physical education teachers still retain their own intentions. Rather than blindly following policies, they maintain their behavior intentions.

During the epidemic, to maintain students’ attendance and improve their health through exercise, many teachers are willing to help students through distance teaching. Therefore, if schools can provide guidance on distance teaching and set up helplines, making the process of implementing distance teaching smoother, it will enhance teachers’ behavioral intentions and effectively promote the use of distance teaching behavior.

From H6 and H7, it is found that gender and age had no moderating effect in the UTAUT theoretical model. Unlike the UTAUT model proposed by Venkatesh et al. (2003), this study found that gender and age do not moderate the distance teaching use intention of physical education teachers. In other words, teachers’ use intentions are not related to gender or age, so there is no need to consider and be concerned with differences between male and female teachers or younger and older teachers. This is likely because during the epidemic, all teachers, regardless of gender or age, were left with no option other than distance teaching methods for their classes. Therefore, gender and age do not affect the intention and behavior of using distance teaching.

From H8, it is revealed that the level of experience moderated the relationship between social influence and the behavioral intention of distance teaching among physical education teachers. This suggests that as teachers gain more experience in distance teaching, they are less influenced by social factors, as they develop their own teaching pace and rhythm over time. Researchers infer that with the accumulation of teaching experience, teachers establish their own effective methods of distance teaching that suit both themselves and their students. Furthermore, experience also moderates the relationship between convenience factors and the behavioral intention of distance teaching. This suggests that as teachers become more experienced, they are less affected by convenience factors in influencing their behavior. With sufficient experience, teachers become adept at handling various hardware and software devices and gradually acquire more teaching alternatives, thereby becoming less dependent on specific conveniences. Although distance teaching for junior and senior high school teachers has only been in practice for less than two years, this study highlights the importance of experience for physical education teachers in distance teaching.

Conclusion and suggestions

Conclusion

The study employed UTAUT to explore the behavioral intention and use behavior of distance teaching of elementary physical education teachers in Taiwan. The results and analysis generated from this study demonstrate that the UTAUT is an appropriate theoretical foundation, resulting in a model with good explanatory power for explaining the distance teaching behavior of physical education teachers.

In the model, performance expectancy, effort expectancy, and social influence all have significant positive effects on the behavioral intention of physical education teachers in distance teaching. Facilitating conditions also positively influence use behavior. Regarding moderation effects, gender and age do not moderate the UTAUT model in this context. Conversely, it was found that the more experienced in distance teaching physical education teachers are, the less they are influenced by social influence and facilitating conditions in their behavioral intention and use behavior in distance teaching. These findings can serve as valuable insights for improving the implementation of physical education distance teaching in the future.

Suggestions

According to the research results, suggestions for teachers and schools are provided as follows.

Suggestions for teachers

The research findings show that of the 364 teachers surveyed, 22 (6.0%) were unfamiliar with distance teaching, 293 (80.5%) had acceptable familiarity, and 49 (13.5%) were quite familiar with it. Moreover, the age range of these teachers varied from 23 to 63 years old. This indicates that, despite the numerous challenges posed by distance teaching, over 90% of elementary and junior high school physical education teachers acquired a conceptual understanding of distance teaching after the government’s implementation of the “suspension of classes without suspension of learning” policy for 2 years. Despite the difficulties presented by distance teaching, teachers exhibited high enthusiasm for teaching and are willing to actively overcome challenges to implement it. However, to further support teachers’ teaching intentions, it is suggested that physical education teachers in elementary and junior high schools actively seek appropriate distance teaching software. Through user-friendly interfaces, the complexity of distance teaching can be reduced. Additionally, increasing the number of cameras in the teaching environment can provide students with clearer demonstrations of correct movements, thereby improving students’ learning effectiveness in distance teaching. When students are satisfied and effective in their learning, it also serves as positive feedback for teachers.

Suggestions for schools

According to the research results, it was found that social influence and convenience significantly impact teachers’ intentions and behaviors regarding distance teaching, especially for physical education teachers with limited experience. If schools can actively intervene and provide assistance, they can effectively address the obstacles that may arise from distance teaching and enhance teachers’ intentions and behaviors. To improve physical education teachers’ competence in distance teaching, schools can organize digital media operation workshops, facilitate peer assistance through intra-school and promote inter-school physical education course design and collaboration communities. These measures will help physical education teachers adapt to distance teaching apps as quickly and effectively as possible. Furthermore, if school administrators can provide physical education teachers with sufficient hardware equipment through information technology experts, and assist in resolving potential video and audio challenges that they may encounter during distance teaching, it will help these unique educators transition into distance teaching more quickly.

Limitations

The study subjects focused on elementary school physical education teachers. However, since the nature of students from different age groups and curriculums differ, the results may be different.

Recommendations for future research

This study investigated the distance teaching behavior of elementary school physical education teachers using the UTAUT model. It was found that factors like performance expectancy, effort expectancy, social influence, and facilitating conditions are all highly correlated with teachers’ distance teaching behavior. However, there are likely other important factors that influence the acceptance of technology, which future research could attempt to identify, to improve the predictive and explanatory power of the model. For example, adding variables like perceived enjoyment, self-efficacy, and satisfaction to the model could improve students’ use of mobile learning behaviors (m-learning) (Shaya, Madani & Mohebi, 2023). Additionally, Raffaghelli et al. (2022) pointed out that the UTAUT model showed a disconfirmation effect between overall acceptance of early warning systems in the pre- and post-usage stages. Therefore, a longitudinal model analysis might reveal the impact of pre-usage expectations on post-usage behavior, which is an area that has been less studied.

Supplemental Information

Supplemental Information 1 Raw data.

a1-a4:performance expectance; b1-b4: effort expectancy; c1-c4:social influence; d1-d4: facilitating condition; e1-e4: behavioral intention; f1-f4: use behavior.

We would like to thank all who participated in the survey for this study.

Additional Information and Declarations

Competing Interests

The authors declare that they have no competing interests.

Author Contributions

Hsiu-Chin Huang conceived and designed the experiments, performed the experiments, analyzed the data, authored or reviewed drafts of the article, and approved the final draft.

Ya-Tzu Kung performed the experiments, analyzed the data, authored or reviewed drafts of the article, and approved the final draft.

Ruey-Rong Huang performed the experiments, analyzed the data, authored or reviewed drafts of the article, and approved the final draft.

Wui-Chiu Mui performed the experiments, analyzed the data, prepared figures and/or tables, authored or reviewed drafts of the article, and approved the final draft.

Yu-Chien Su conceived and designed the experiments, prepared figures and/or tables, and approved the final draft.

Human Ethics

The following information was supplied relating to ethical approvals (i.e., approving body and any reference numbers):

National Chiayi University.

Data Availability

The following information was supplied regarding data availability:

The raw data is available in the Supplemental File.

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
