# Peer review of "Assessment of physical education teachers’ use of distance teaching behavior under the influence of the COVID-19 pandemic"

_PeerJ, doi:10.7717/peerj.18743_

## Round 0.1 · original submission · Major Revisions

Thank you for your submission. The reviewers have identified a number of concerns that must be addressed.

Also, provide a stronger justification of the study. Include information on ethics approval. How were the participants were recruited? Provide more details about the instrument used, what modifications were made from the original as well as the language used. Recommendations should not be in the Conclusion.

The writing would benefit from proofreading and editing by a native speaker.

Reviewer 1 ·

Basic reporting

The introduction identifies the relevance of research in the context of the Covid 19 pandemic and the role of digital technologies in optimizing the physical education teaching process in schools. The 3 categories of analyzed variables (4 independent, 3 moderating and 2 dependent) are clearly described, within the UTAUT model (model also applied in other fields). The manuscript is well structured in sections, the tables and graphs inserted at the end of the article correctly summarize the results of the study. The references to which the results are reported are useful and consistent with the research theme.

Experimental design

The very large number of main and secondary hypotheses gives this study a high level of complexity. The statistical procedures applied are in accordance with the type of investigation carried out. The 7-level/point Likert scale for the 6-component/24-item structured questionnaire ensures an accurate measurement of the opinions of the teachers in the studied sample. Additional documents confirm compliance with the rules of research involving human subjects/Declaration of Helsinki.

Validity of the findings

The authors identified the common aspects and the differences found, by comparing them with the results of other similar studies. The investigation offers important ideas related to highlighting the factors/variables that favorably influence the process of teaching physical education in the online environment. The conclusions, recommendations and new research directions identified are correctly associated with the results presented and the hypotheses formulated.

Additional comments

1. Abstract Section: It would be useful to mention what PLS statistical analyzes (Partial least squares regression) means. Method: first independent variable/ behavioral intention (BE)? The same for the results: The results found PE, EE and SI had significant influences on BE and FC and BE had significant influences on UB. I think it needs to be changed (BI).
2. Hypotheses H6d, H7d and H8d (lines 159-160, 166-167, 176-177): The statement of these hypotheses shows that the variables gender, age and experience moderate the effect of facilitating conditions on teachers' intention to engage in distance teaching . BI (behavioral intention) is the first dependent variable. Figure 1 shows how the variables gender, age and experience moderate the effect of facilitating conditions on Use Behavior/UB (the second dependent variable analyzed). The same relationships between variables are also mentioned in the results section: lines 322-323, 334-335, 348-349.
3. Descriptive statistics of the respondents (lines 221-223): Maybe this data could be included in a table..... it's just a suggestion. It is not clear whether all participants teach physical education in urban areas, or there are also cases that teach in rural areas.
4. Can you specify the time frame in which you investigated the sample of teachers?
5. Tables 3 and 4 (Horizontal variables): What do HM and PV define/mean? In Note these variables do not exist. The vertical variables: US1, US2, US3, US4 and US? I think UB (use behavior) is analyzed.

·

Basic reporting

Review the manuscript for grammatical errors and awkward phrasings, possibly with the assistance of a professional editor.

Expand the literature review to include more recent studies and provide a more comprehensive background.

Improve the flow between sections, especially the transition from the introduction to the methodology.
Ensure all figures and tables are of high quality and integrate them better into the text.
Provide more detailed explanations of the raw data and analysis processes.

Experimental design

The manuscript presents a study utilizing the Unified Theory of Acceptance and Use of Technology (UTAUT) model to explore physical education teachers' use of distance teaching during the COVID-19 pandemic. While the study's design is generally sound, there are a few areas where improvements could be made:

The research question is well-defined, and the study aims to fill a knowledge gap regarding the application of distance teaching in physical education. However, the objectives could be more clearly articulated. Clearly stating the specific goals and expected outcomes at the end of the introduction or in a separate section would provide a better roadmap for the study.

The study included a sample of 364 valid responses from physical education teachers. While the sample size is adequate, there is limited information on the sampling method. Details on how participants were selected, any inclusion or exclusion criteria, and efforts to ensure a representative sample would enhance the study's validity and generalizability.

The use of Partial Least Squares (PLS) statistical analysis is appropriate for testing the relationships among variables. However, the description of the methodology could be more comprehensive. For example, the process of developing and validating the questionnaire items, including any pilot testing or expert review, should be elaborated upon to establish the instrument's reliability and validity.

The study includes moderating variables such as gender, age, and experience. However, there could be other relevant factors that were not considered, such as the level of technological proficiency or the availability of technological resources. Discussing these potential confounding variables and how they were controlled (or not) would provide a more complete picture of the study's design.

The study mentions that the survey was approved by National Chiayi University and that participants consented. However, it would be beneficial to provide more details about the ethical considerations, such as how participants' anonymity and data privacy were ensured and whether there were any incentives for participation.

Suggested improvements include clearly stating the research objectives and expected outcomes, providing more detail on the sampling method, including any inclusion/exclusion criteria and efforts to ensure a representative sample, elaborating on the development and validation process of the questionnaire items, considering other potential control variables and how they were managed, and providing more information on the ethical considerations and measures taken to protect participants' data and privacy.

Validity of the findings

The findings of the study are generally valid, but there are some aspects that could be strengthened to enhance the credibility and robustness of the conclusions.

The study effectively uses the Unified Theory of Acceptance and Use of Technology (UTAUT) model to analyze the factors influencing physical education teachers' use of distance teaching. The statistical methods employed, including Partial Least Squares (PLS) analysis, are appropriate for the research questions and allow for a detailed examination of the relationships between variables. However, there are areas where the validity of the findings could be further substantiated.

One potential issue is the reliance on self-reported data, which can introduce biases such as social desirability or recall bias. The authors should discuss the limitations of self-reported measures and consider how these might impact the findings. Additionally, the study could benefit from including objective measures or triangulating the data with other sources, such as classroom observations or interviews, to provide a more comprehensive view of the teachers' behavior and attitudes.

The sample size of 364 participants is adequate, but the representativeness of the sample is not clearly addressed. Information on the demographic and professional characteristics of the sample, such as geographical distribution, years of teaching experience, and types of schools, would help determine the generalizability of the findings. Without this information, it is difficult to assess whether the results can be extrapolated to a broader population of physical education teachers.

The study's findings are supported by the data analysis, with statistically significant relationships observed for most of the hypothesized effects. However, the authors should discuss the practical significance of these findings, not just the statistical significance. For example, while certain factors like performance expectancy and social influence were found to significantly impact behavioral intention, the practical implications of these effects should be explored. How do these factors translate into actual changes in teaching practice? What recommendations can be made for policymakers or educators based on these findings?

The authors should also consider the potential confounding variables that might influence the results. While the study accounts for variables like gender, age, and experience, other factors such as prior experience with technology or access to resources could also play a role. A more thorough discussion of these potential confounders and their impact on the results would add depth to the analysis.

In conclusion, while the findings are generally valid and supported by the data, the study could benefit from a more detailed discussion of the limitations, practical significance, and potential confounding variables. These additions would strengthen the credibility and applicability of the conclusions.

Additional comments

none

---

## Round 0.2 · Minor Revisions

The reviewers have highlighted the need for more clarity and detail in your methodology and discussion.

Reviewer 1 ·

Basic reporting

The manuscript has been completed in accordance with the suggestions submitted.

Experimental design

Everything is ok.

Validity of the findings

Everything is ok.

Additional comments

The authors have modified the manuscript in accordance with the suggestions submitted.

·

Basic reporting

The manuscript uses professional and clear English, making it accessible to a wide audience. However, there are minor grammatical errors and awkward phrasing in the introduction (e.g., "switched from established in-person teaching to distance teaching was quite challenging"). These could be improved for smoother reading.
Suggestion: A professional proofreader or language editing service could enhance the clarity and flow.

The study provides a sufficient background on the Unified Theory of Acceptance and Use of Technology (UTAUT) model and its application in diverse fields. However, more discussion on gaps in previous research specific to physical education and distance learning during COVID-19 would strengthen the rationale.
Suggestion: Expand the introduction to highlight specific challenges or gaps in the literature related to physical education and distance teaching.

The article adheres to PeerJ's structure standards, and the figures and tables are relevant, well-labeled, and described. However, the hypothesized model in Figure 1 could benefit from clearer labeling of constructs for ease of interpretation by non-specialist readers.
Suggestion: Ensure that all constructs in figures are explicitly labeled and described in the figure legends.

Raw data has been shared, fulfilling PeerJ's policy. Ensure that metadata is sufficiently descriptive for future use by other researchers.
Suggestion: Add more detailed metadata to the raw data file for greater transparency.

The manuscript presents relevant results that align with the stated hypotheses, providing a self-contained narrative. No significant issues in this regard.

Experimental design

The study fits well within the journal's aims and scope, addressing the impact of distance teaching on physical education during the COVID-19 pandemic. The research question is clearly defined, relevant, and meaningful. It identifies a gap in understanding how physical education teachers adapt to distance teaching, which is particularly relevant in a post-pandemic context.
Suggestion: Strengthen the discussion on how the research addresses specific gaps, such as limited studies on UTAUT's application to physical education.

Ethical considerations are adequately addressed, with informed consent obtained from participants and adherence to local research regulations. The study meets ethical standards for research involving human participants.
Suggestion: Include details on how anonymity and data confidentiality were maintained for respondents to enhance transparency.

The methodology is robust, employing the UTAUT model and Partial Least Squares Structural Equation Modeling (PLS-SEM), which are appropriate for exploring behavioral intention and use behavior. However, the study could benefit from more discussion on why PLS-SEM was chosen over other statistical methods.
Suggestion: Provide a brief justification for selecting PLS-SEM in terms of its advantages for analyzing complex models with latent variables.

The methods section is comprehensive, with detailed descriptions of participant recruitment, survey administration, and data analysis. The use of a validated questionnaire adapted for the physical education context is commendable. However, the process of adapting the UTAUT questionnaire could be described in greater detail, especially the expert validation process.
Suggestion: Elaborate on the steps taken to modify and validate the UTAUT questionnaire for the physical education context, including how expert input was incorporated.

The sample size of 364 valid responses is adequate for PLS-SEM analysis. The study includes a diverse range of physical education teachers from various regions, enhancing generalizability. However, there is limited discussion on how the sample represents the broader population of physical education teachers.
Suggestion: Include a brief discussion on the representativeness of the sample and potential limitations in generalizing the findings.

The study provides sufficient detail for replication, including the statistical tools used and the questionnaire items. Sharing the raw data further supports reproducibility.
No additional suggestions.

Validity of the findings

While impact and novelty are not directly assessed, the study addresses an important topic—distance teaching in physical education during the COVID-19 pandemic. The findings provide valuable insights into teachers' adoption of technology using the UTAUT framework, contributing to the literature.
Suggestion: Emphasize how the findings extend current knowledge or provide new practical implications for physical education and distance teaching.

The rationale for the study is well-articulated, and the findings are positioned to encourage meaningful replication in other educational contexts. However, further discussion on how the findings could be applied to different educational settings or disciplines would strengthen the paper.
Suggestion: Include a brief paragraph in the discussion section addressing how the methodology and findings could be replicated or extended to other contexts.

The underlying data appear robust and statistically sound, with appropriate use of PLS-SEM for hypothesis testing. The results are clearly presented and controlled for potential biases, such as multicollinearity.
No major issues observed in data validity.:

The conclusions are well-stated, linked to the research questions, and supported by the results. They effectively summarize the key findings and provide actionable recommendations for teachers and schools. However, the limitations section could be expanded to address broader contextual or methodological constraints.
Suggestion: Enhance the limitations section to include potential biases, such as reliance on self-reported data and the specific cultural context of the study.

The raw data has been provided, supporting the reproducibility of the results. However, the manuscript could benefit from explicitly mentioning how external researchers might access and use the dataset for future work.
Suggestion: Add a statement on how the raw data aligns with the journal's data-sharing policy and encourage its use for further research.

Additional comments

The manuscript provides a comprehensive study on the use of distance teaching among physical education teachers. The findings are valuable and well-presented, with practical implications for education policy and practice. However, incorporating additional references can further enrich the discussion and align the study with recent literature.

---

## Round 0.3 · accepted · Accept

Thank you for your revised submission. I am satisfied that you have addressed the remaining concerns of the reviewer, and am happy to accept your paper for publication.

·

Basic reporting

I Would like to thank the authors for addressing all the comments, and recommend the publications of the paper as is.

Experimental design

I Would like to thank the authors for addressing all the comments, and recommend the publications of the paper as is.

Validity of the findings

I Would like to thank the authors for addressing all the comments, and recommend the publications of the paper as is.

Additional comments

I Would like to thank the authors for addressing all the comments, and recommend the publications of the paper as is.